# The ‘Ins and Outs’ of Early Preclinical Models for Brain Tumor Research: Are They Valuable and Have We Been Doing It Wrong?

**DOI:** 10.3390/cancers11030426

**Published:** 2019-03-25

**Authors:** Ola Rominiyi, Yahia Al-Tamimi, Spencer J. Collis

**Affiliations:** 1Department of Neurosurgery, Sheffield Teaching Hospitals NHS Foundation Trust, Sheffield S10 2JF, UK; y.al-tamimi@sheffield.ac.uk; 2Department of Oncology & Metabolism, The University of Sheffield Medical School, Sheffield S10 2RX, UK; s.collis@sheffield.ac.uk; 3NIHR Sheffield Biomedical Research Centre, Sheffield Teaching Hospitals NHS Foundation Trust, Sheffield S10 2JF, UK

**Keywords:** pre-clinical, research models, 3D models, brain tumors, cancer stem cells, glioblastoma, DNA damage response (DDR)

## Abstract

In this perspective, we congratulate the international efforts to highlight critical challenges in brain tumor research through a recent Consensus Statement. We also illustrate the importance of developing more accurate and clinically relevant early translational in vitro brain tumor models—a perspective given limited emphasis in the Consensus Statement, despite in vitro models being widely used to prioritize candidate therapeutic strategies prior to in vivo studies and subsequent clinical trials. We argue that successful translation of effective novel treatments into the clinic will require investment into the development of more predictive early pre-clinical models. It is in the interest of researchers, clinicians, and ultimately, patients that the most promising therapeutic candidates are identified and translated toward use in the clinic. Highlighting the value of early pre-clinical brain tumor models and debating how such models can be improved is of the utmost importance to the neuro-oncology research community and cancer research more broadly.

## 1. Introduction

The recent Consensus Statement ‘Challenges to curing primary brain tumors [1]’, produced by a Cancer Research UK convened panel of international experts, represents an important ‘call-to-arms’ to the neuro-oncology research community and should be congratulated for highlighting areas worthy of investment to improve patient outcomes. Agreement exists within the cancer research community that more accurate genetically engineered mouse models (GEMMs) and orthotopic patient-derived xenografts (PDXs), along with greater sharing of these resources, will help translate new treatments into the clinic. However, it is important to consider that, in some contexts, these in vivo models may only be as successful as the therapeutic strategies they are used to evaluate. Consequently, greater emphasis should also be directed toward enhancing and validating early pre-clinical in vitro culture systems used to model brain tumors. 

## 2. The Value of Early Pre-Clinical Models in Brain Tumor Research

Using the example of targeting the DNA damage response (DDR) [2], effective future treatment strategies may need to incorporate novel therapeutic combinations (such as combined PARP and ATR inhibition with Olaparib and AZD6738, respectively) to overcome the extensive intra-tumoral heterogeneity brain tumors can exhibit, and functional redundancy governed by the interconnectedness of many biological processes [3]. At least 19 novel drugs that target the DDR in cancer are currently approved or in clinical trials [2]. Conservative estimates provide at least 147 potential two-drug or 939 three-drug DDR inhibitor combinations (excluding strategies that combine drugs of the same class). In the quest to improve patient outcomes, all of these combinations may warrant evaluation, however, testing every combination using rodent models would be impractical and potentially unethical. Therefore, robust validation of biologically relevant in vitro models that are able to recapitulate key disease features and predict clinical response may represent one of the most urgent and important challenges facing our community.

## 3. A Need to Validate and Contrast In Vitro Models

Whilst organoids represent a promising 3D culture system highlighted in the Consensus Statement, other clinically relevant 3D co-culture systems do exist [4,5], and the relative value of each to prospectively determine which therapies extend survival for brain tumor patients in Phase III clinical trials remains unclear. Furthermore, the Consensus Statement alludes to the blood–brain barrier (BBB) representing a key obstacle to the efficient delivery of chemotherapy and novel therapeutics to brain tumors [1]. In recent years, numerous advances in modeling the BBB have been made including microfluidic ‘BBB on a chip’ systems [6] and multi-cellular BBB spheroids, based on non-adherent co-culture of human astrocytes, pericytes, and brain endothelial cells [7]. Refining the ability of in vitro models such as these to reproduce in vivo selectivity of both the BBB and blood-tumor barrier remains critically important to accurately predict the ability of novel therapeutics to reach brain tumors. Increasingly, the ‘gold-standard’ assessment of each should perhaps be based on the ability to predict tumoral or peri-tumoral drug concentrations and intended target effects in Phase 0 clinical trials, rather than animal studies. To prioritize novel therapies for the future, many researchers would benefit from the clear validation of which in vitro BBB models best predict drug delivery in human subjects. Greater investment in these areas is crucial to avoid the loss of talent within the research community and ensure candidate therapies with the greatest likelihood of success are selected for in vivo studies and subsequent clinical trials.

## 4. Toward Early Translational Models of Post-Surgical Disease

Additionally, worthy of consideration are the neurosurgical features of deriving human brain tumor cells used in the majority of pre-clinical models, whether in vitro (2D, 3D, and organoid culture systems) or in vivo (PDXs). Generally, these cells come from the resected tumor taken out at surgery, even though the infiltrative residual cells left inside patients after surgery are responsible for disease progression. For example, considering glioblastoma, which is characterized by extensive spatial and temporal heterogeneity [8,9], it should not be assumed that the more invasive cells left behind after surgery respond to treatment in an identical fashion to those cells that are removed. Consequently, greater emphasis should be placed on the incorporation of typical residual cells into patient-derived pre-clinical models. Our group and others are currently working on such models of clinically relevant post-surgical residual disease, which we envisage will be valuable to the research community and help prioritize the most promising novel therapies in the context of current multi-modal treatment.

## 5. Conclusions

Highlighting the value of early pre-clinical brain tumor models and debating how such models can be improved is essential to enable the identification and translation of novel treatment strategies that are most likely to be successful in clinical trials for patients with brain tumors. In cancer research, developing more clinically relevant in vitro models to better prioritize novel therapies worthy of further investigation is critical. The translational ‘pipeline’ may only ever be as successful as those treatments which enter it.

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
