# Peer review of "The ‘Ins and Outs’ of Early Preclinical Models for Brain Tumor Research: Are They Valuable and Have We Been Doing It Wrong?"

_cancers, 2019, doi:10.3390/cancers11030426_

Reviewer 1 Report

In this manuscript, the authors highlighted the value of early pre-clinical models in brain tumor research using the example of targeting the DNA damage response, and discussed the need to validate and contrast in vitro models. Overall, it is written well, and the conclusions are clear. 

Author Response

Response to Reviewer 1 Comments

Point 1: In this manuscript, the authors highlighted the value of early pre-clinical models in brain tumor research using the example of targeting the DNA damage response, and discussed the need to validate and contrast in vitro models. Overall, it is written well, and the conclusions are clear.

Response 1: We agree with Reviewer 1 and are very pleased to receive this positive feedback.

Reviewer 2 Report

Nice commentary. 

Page 2 line 64. The statement ' The equivalence of these spatially divergent subpopulations cannot be assumed.'  This was not clear - not sure if I understand the point of this section. Consider revising. 

I think another point that ought to be raised is that the ability of these drugs, chemicals and small molecule inhibitors to pass through the blood brain barrier (BBB) should be considered prior to costly pre-clinical in vivo studies.  Not sure it the authors want to mention the current 3-D models of the BBB that have recently been published.  

Author Response

Response to Reviewer 2 Comments

Point 1: Nice commentary.

Response 1: We are very pleased to receive this positive feedback and do feel this this perspective/commentary article would be of interest to the cancer research community.

Point 2: Page 2 line 64. The statement ' The equivalence of these spatially divergent subpopulations cannot be assumed.' This was not clear - not sure if I understand the point of this section. Consider revising.

Response 2: We have taken this feedback on board, and in retrospect agree that the message could have been conveyed more clearly. Our intention in this section was to describe that glioblastoma are known to exhibit high levels of heterogeneity (genetic and functional) within the tumour, consequently we should not assume that cells that have migrated futher away from the tumour (to infiltrate adjacent brain) are identical to those which remain inside the tumour. In view, of the insightful comments by Reviewer 2 we have revised this section to add clarity (Page 2 lines 81-83).

Point 3: I think another point that ought to be raised is that the ability of these drugs, chemicals and small molecule inhibitors to pass through the blood brain barrier (BBB) should be considered prior to costly pre-clinical in vivo studies. Not sure it the authors want to mention the current 3-D models of the BBB that have recently been published.

Response 3: We feel this is also an insightful and constructive suggestion. Consequently, we have added discussion of the value of early pre-clinical BBB models (Page 2 lines 62-73). Within this discussion we also suggest greater use of more clinically relevant ways to assess the predictive value of such models would be helpful to pre-clinical researchers.